

# Correcting a Systematic Bias in an Ocean Drilling Project Site 882 Alkenone Sea Surface Temperature Record

Joseph B. Novak[1], Rocío P. Cabellero-Gill[2], Timothy D. Herbert[3], Harry J. Dowsett[4], Alfredo Martínez-García[5]

[1]Ocean Sciences Department, University of California, Santa Cruz, CA, U.S.A.
[2]Department of Atmospheric, Oceanic and Earth Sciences, George Mason University, Fairfax, VA, U.S.A.
[3]Department of Earth, Environmental and Planetary Sciences, Brown University, Providence, RI, U.S.A.
[4]U.S. Geological Survey, Florence Bascom Geoscience Center, Reston, VA, U.S.A.
[5]Max Plank Institute for Chemistry, Mainz, Germany

*Correspondence to*: Joseph B. Novak (jobnovak@ucsc.edu)

**Abstract.** Reconstructions of sea surface temperature (SST) in the geologic record are fundamental to our understanding of Earth's climate history and the evaluation of Earth's climate sensitivity to greenhouse gas forcing. SSTs are reconstructed with a variety of methods, including alkenone biomarker lipids produced by certain coccolithophore algae. One such alkenone SST reconstruction from the subpolar northwest Pacific Ocean Drilling Project (ODP) Site 882 has played a large

role in shaping the paleoclimate science community's view of global climate warmth during the Late Pliocene (3.6–2.6 million years ago) and the subsequent cooling that characterized the intensification of Northern Hemisphere Glaciation (Haug, 1995; Haug et al., 2005; Martínez-Garcia et al., 2010). Here, using published data from ODP Site 882 (Studer et al., 2012) and nearby Site 883 (Novak et al., 2024), we demonstrate that the long alkenone SST at ODP Site 882 systematically reports an amplified range of absolute SST values, including maximum SSTs 2–4°C warmer than the more recently

generated data. We suggest that the difference between these datasets is a result of the gas chromatography chemical ionization mass spectrometry (GC-CI-MS) analytical method used by Haug (1995), which is consistent with known challenges with this method (Chaler et al., 2000, 2003). We show that alkenone SST estimates derived from the gas chromatography flame ionization detector (GC-FID) method at Sites 882 and 883 have qualitatively similar trends but are systematically offset in their absolute values from the data first reported from ODP Site 882 (Haug, 1995; Haug et al., 2005;

Martínez-Garcia et al., 2010). While this finding does not invalidate the conclusions of the original studies, it does strongly suggest that absolute values derived from published alkenone SST estimates from ODP Site 882 are not suitable for evaluating Earth System Model climate simulations. As an alternative, we present a corrected ODP Site 882 alkenone SST dataset that more closely agrees with the published GC-FID data, albeit with larger uncertainties in the reconstructed SSTs.



## 1 Introduction

Paleoclimate data are the only means of evaluating the ability of Earth System Models to simulate warm climates under boundary conditions drastically different than those captured by the recent historical record (Tierney et al., 2020). Typically, this evaluation comes in the form of direct comparisons between model-simulated fields of sea surface temperature (SST) and geological proxy observations of SST (e.g., Haywood et al., 2016, 2020). These studies have identified shortcomings in climate model simulations, particularly the tendency of climate models to emulate a steeper latitudinal temperature gradient

than is indicated by the geological data across several intervals of past warm climate (Burls et al., 2021; Haywood et al., 2016; Huber and Caballero, 2011).

Fundamentally, these data-model comparison efforts rely upon reliable geologic proxy SST estimates. One such proxy is alkenones – algal biomarkers made by certain coccolithophore algae that are sensitive to seawater temperature (Brassell et al., 1986; Marlowe et al., 1984, 1990). The alkenone paleothermometer is based upon the relative abundance of tri-

unsaturated 37-carbon methyl alkenones ($C_{37:3}Me$) and di-unsaturated 37-carbon methyl alkenones ($C_{37:2}Me$), which form the $U^{K'}_{37}$ ratio (Prahl and Wakeham, 1987).

$$U^{K'}_{37} = \frac{C_{37:2}Me}{C_{37:3}Me + C_{37:2}Me} \tag{1}$$

The relationship between the $U^{K'}_{37}$ ratio and SST has been calibrated by longstanding efforts to analyse alkenones in marine surface sediments (Conte et al., 2006; Müller et al., 1998; Novak et al., 2022; Prahl et al., 2010; Rosell-Melé et al., 1995;

Sikes et al., 1991; Tierney and Tingley, 2018). Alkenones are commonly measured by gas chromatograph flame ionization detector (GC-FID) (Longo et al., 2013; Villanueva et al., 1997; Villanueva and Grimalt, 1997; Zheng et al., 2017). A more sensitive gas chromatograph positive chemical ionization mass spectrometry technique has been used in sites with very low alkenone concentrations (Rosell-Mele et al., 1995). However, subsequent studies showed that this method can be hindered by complex concentration-dependent nonlinear effects on the chemical ionization of alkenones that impact the $C_{37:3}Me$ and

$C_{37:2}Me$ ketones differently (Chaler et al., 2000, 2003), leading to systematic offsets in SSTs reconstructed from the two techniques. Because there is not an authentic alkenone laboratory standard, resolving these method-dependent nonlinear differences between the GC-FID and GC-CI-MS methods requires instrument-specific calibration of GC-CI-MS $U^{K'}_{37}$ values to those determined on GC-FID (Chaler et al., 2000, 2003).

Here, we revisit the 5.7-million-year (Ma) alkenone SST record from the subpolar northwest Pacific Ocean Drilling Project

(ODP) Site 882, which was determined by the GC-CI-MS method (Haug, 1995). We show that, as expected, the ODP Site 882 alkenone SST record is systematically warmer than alkenone SSTs in samples of the same age from ODP Site 882 (Studer et al., 2012) and nearby ODP Site 883 determined by GC-FID (Novak et al., 2024). This systematic offset is consistent with the concentration-dependent effects on the chemical ionization of alkenones (Chaler et al., 2000, 2003) for





the range of trace alkenone concentrations reported at ODP Site 882 (Haug et al., 2005; Studer et al., 2012). Here, we

propose a corrected version of the original ODP Site 882 alkenone record that accounts for additional uncertainties in reconstructed SSTs based on propagating the errors in the relationship between the ODP Site 882 GC-CI-MS and GC-FID alkenone data.

## 2 Methods

### 2.1 Study Sites and Published Alkenone SSTs

Here we analysed published alkenone SST datasets from ODP Site 882 and ODP Site 883, located 49 nautical miles to the north of Site 882 (Rea et al., 1993a, b) (Fig. 1). The original ODP Site 882 alkenone SST record was determined by GC-CI-MS (Haug, 1995), and was subsequently published by Haug et al. (2005) and Martínez-Garcia et al. (2010). The potential concentration-dependent nonlinear differences between GC-FID and GC-CI-MS $U^{K'}_{37}$ determinations identified by Chaler et al. (2000, 2003) and their impacts on absolute SST values were not discussed in these publications, which focused mainly on

the analysis of reconstructed SST trends. More recently, the ODP Site 882 alkenone SST record of Studer et al. (2012) and the ODP Site 883 alkenone SST record of Novak et al. (2024) were determined by GC-FID (Villanueva et al., 1997; Villanueva and Grimalt, 1997), allowing the assessment of potential biases in the original dataset from ODP Site 882 (Haug, 1995). The alkenone SST estimates we compare here were generated from the BAYSPLINE alkenone SST calibration of Tierney and Tingley (2018).

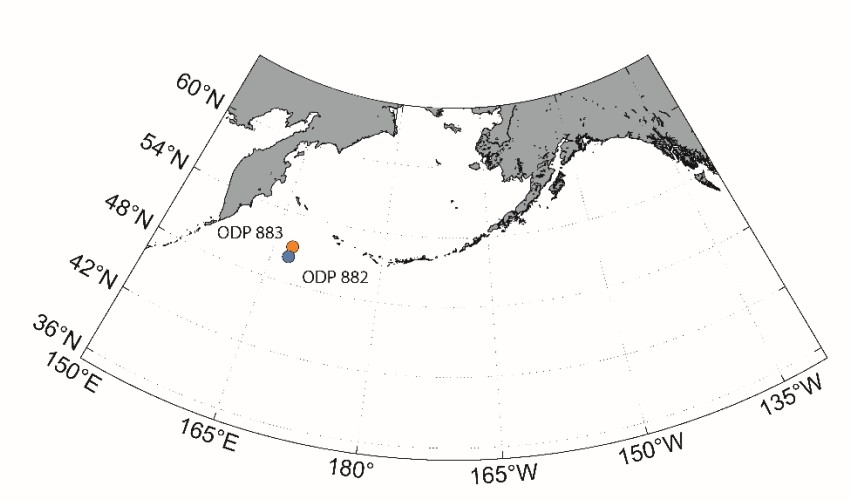


**Figure 1: Location of Ocean Drilling Program Leg 145 Sites 882 and 883. Map generated with M_Map for MATLAB** (Pawlowicz, 2020)**.**



## 2.2 Concentration-dependent differences in GC-CI-MS and GC-FID $U^{K'}_{37}$ SST Determinations

Chaler et al. (2003) derived a mathematical expression of $U^{K'}_{37}$ as measured on GC-FID as a function of GC-CI-MS measurements of alkenone concentration:

$$U^{K'}_{37} = \frac{(c_{2M}C^2_{37:2}+ b_{2M}C_{37:2}+ a_{2M})}{(c_{2M}C^2_{37:2}+ b_{2M}C_{37:2}+ a_{2M}+ c_{3M}C^2_{37:3}+ b_{3M}C_{37:3}+ a_{3M})} \qquad (2)$$

Here, $a_{2M}$, $a_{3M}$, $b_{2M}$, $b_{3M}$, $c_{2M}$, and $c_{3M}$ are curve fitted constants reported by Chaler et al. (2003). These constants are instrument-specific (Chaler et al., 2003). We utilized the constants from GC-CI-MS "Instrument A" of Chaler et al. (2003) to calculate theoretical GC-FID-equivalent $U^{K'}_{37}$ values from theoretical GC-CI-MS alkenone measurements that we then transformed into SST estimates via BAYSPLINE to facilitate comparison between the GC-FID and GC-CI-MS alkenone SST datasets from ODP Sites 882 and 883 (Table 1). "Instrument A" of Chaler et al. (2003) was chosen over "Instrument B" because the range of analyte masses measured by "Instrument A" better corresponded to the range of $U^{K'}_{37}$ values in the alkenone SST records from ODP Sites 882 and 883 (Novak et al., 2024; Studer et al., 2012). We varied mass of each theoretical alkenone analyte between 5 and 30 ng on column to mimic the "trace" quantities of alkenones reported by Haug et al. (2005) because no specific alkenone concentrations were reported alongside their SST determinations. For example, a low $U^{K'}_{37}$ value was represented as 5 ng of the $C_{37:2}$Me alkenone and 30 ng of the $C_{37:3}$Me alkenone. These analyte masses are slightly below to slightly above the lower bound of the range of acceptable analyte mass for $U^{K'}_{37}$ determinations by GC-FID as documented by Villanueva and Grimalt (1996).

| Constant | Value |
|:---:|:---:|
| $a_{2M}$ | 1.9468 |
| $a_{3M}$ | 1.0065 |
| $b_{2M}$ | 1.2836 |
| $b_{3M}$ | 7.388 |
| $c_{2M}$ | 0.0165 |
| $c_{3M}$ | 0.0236 |

Table 1: constants used to calculate GC-FID-equivalent $U^{K'}_{37}$ values from theoretical GC-CI-MS alkenone measurements (Chaler et al., 2003).

## 2.3 Correcting the ODP Site 882 Alkenone SST Record

Because we lack instrument-specific information about the mass-dependent ionization efficiency of the $C_{37:3}$Me and $C_{37:2}$Me alkenones on the GC-CI-MS used by (Haug, 1995) to generate their $U^{K'}_{37}$ data from ODP Site 882, we cannot use the equations of Chaler et al. (2003) to correct their record. We propose a simple correction based on the linear correlation between alkenone SST estimates in adjacent samples at ODP Site 882 generated by GC-CI-MS and GC-FID (Haug, 1995; Studer et al., 2012) (Fig. 3a, adj. $r^2$ = 0.66, p < 0.001, df = 18, residual standard error = 2.5°C SST, Eq. 3).



$$GC\text{-}FID\ ODP\ 882\ SST\ =\ GC\text{-}CI\text{-}MS\ ODP\ 882\ SST * 0.366(\pm0.06°C) + 5.9(\pm0.7°C) \qquad (3)$$

We applied equation 3 to the median BAYSPLINE alkenone SST estimates generated from the Haug (1995) dataset. We then propagated the additional uncertainty to calculate a 95% confidence interval for the resulting median SST estimates by
adding in quadrature the residual standard error multiplied by 1.96 to the upper and lower range of the 95% SST confidence interval returned by BAYSPLINE.

$$95\%\ CI\ lower\ bound\ =\ GC\text{-}FID\ ODP\ 882\ SST\ -$$
$$\sqrt{(BAYSPLINE\ median\ SST - BAYSPLINE\ SST\ 95\%\ CI\ lower\ bound)^2 + (1.96 * Eq.3\ residual\ standard\ error)^2}$$
$$(4)$$

$$95\%\ CI\ upper\ bound\ =\ GC\text{-}FID\ ODP\ 882\ SST\ +$$
$$\sqrt{(BAYSPLINE\ SST\ 95\%\ CI\ upper\ bound - BAYSPLINE\ median\ SST)^2 + (1.96 * Eq.3\ residual\ standard\ error)^2}$$
$$(5)$$

Because BAYSPLINE generates a unique 95% confidence interval for each sample, this calculation was applied to the BAYSPLINE output for each sample individually.




## 3 Results and Discussion

### 3.1 Comparison of Alkenone SST Records

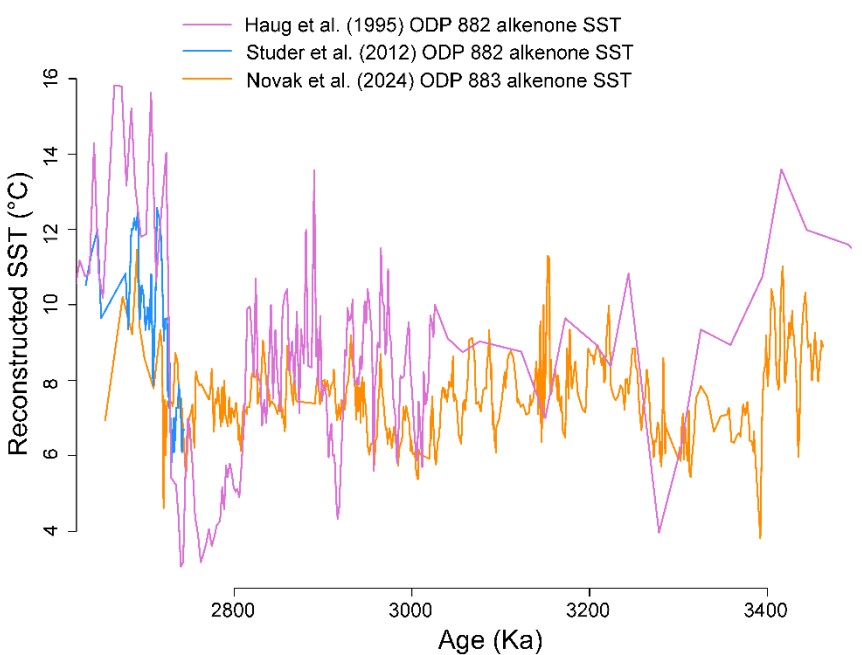

**Figure 2: Timeseries of alkenone sea surface temperature reconstructions from ODP Sites 882 and 883. Note that the SSTs shown are the median of the BAYSPLINE output.**

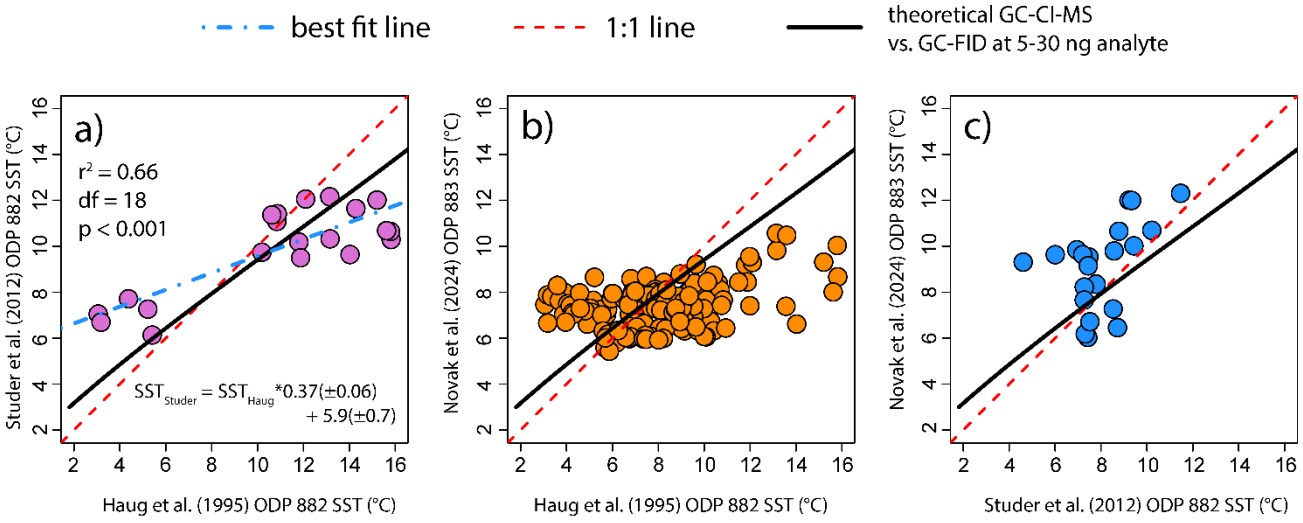




**Figure 3: Alkenone sea surface temperature reconstructions from ODP Sites 882 and 883 interpolated to show differences in absolute SST estimates in samples of similar age. The black line was calculated from $U^{K'}_{37}$ values derived from the equations reported for GC-CI-MS "instrument A" in Chaler et al. (2003) (see 2.2) converted to SST by BAYSPLINE (Tierney and Tingley, 2018). In all cases, the more highly time resolved record was interpolated onto the lower resolution record.**

Comparison of the reconstructed SST timeseries from ODP Sites 882 and 883 shows that the record of Haug et al. (2005) generated by GC-CI-MS is often warmer by approximately 2–4°C than the records of Studer et al. (2012) and Novak et al. (2024) except for a few short intervals where all three records reconstruct SSTs on the order of 8°C (Fig. 2). We also note two intervals where the Haug et al. (2005) alkenone SST record is colder than the records of Studer et al. (2012) and Novak et al. (2024), also by 2–4°C (Fig. 2). Point-to-point comparison of interpolated reconstructed SSTs from the three records is

characterized by systematically warmer maximum SSTs and systematically colder minimum SSTs in the Haug et al. (2005) record that converge with SSTs reconstructed by Studer et al. (2012) and Novak et al. (2024) at the 6–10°C SST range (Fig. 3). Consequently, the range of SSTs reconstructed by Haug et al. (2005) is greater than that reconstructed by Studer et al. (2012) and Novak et al. (2024) (Fig. 2). We also note that the SSTs reconstructed by Studer et al. (2012) and Novak et al. (2024), both determined by GC-FID, do not appear to be systematically offset (Fig. 2 & 3c).

The differences we observe between the GC-CI-MS determined alkenone SST record of Haug et al. (2005) and the GC-FID determined alkenone SST records of Studer et al. (2012) and Novak et al. (2024) are typical of the concentration-dependent nonlinear differences between GC-FID and GC-CI-MS $U^{K'}_{37}$ determinations identified by Chaler et al. (2000, 2003) (Fig. 3). Specifically, $U^{K'}_{37}$ values determined by GC-CI-MS and GC-FID broadly converge around $U^{K'}_{37}$ values of approximately 0.3–0.4 (~6–10°C SST) and are systematically warmer at $U^{K'}_{37}$ values > 0.4 and systematically colder at $U^{K'}_{37}$ values < 0.3

(Chaler et al., 2000, 2003) (Fig. 3). These offsets in $U^{K'}_{37}$, and therefore reconstructed SSTs, are instrument-specific (see Fig. 3 of Chaler et al., 2003), complicating the correction of SSTs reported by Haug et al. (2005) for this effect without substantial investigation of the ionization efficiency of the GC-CI-MS used in that study. Likewise, although the range of affected $U^{K'}_{37}$ values, and therefore reconstructed SSTs, reported by Chaler et al. (2000, 2003) are broadly true, they likely do not represent the exact response of alkenones, and therefore $U^{K'}_{37}$ values, for the specific instrument used by Haug et al.

(2005). Nevertheless, the equations reported by Chaler et al. (2003) approximate the systematic offset between the GC-CI-MS and GC-FID datasets (Fig. 3), particularly when comparing the SSTs of Haug et al. (2005) to those of Studer et al. (2012) (Fig. 3a, black line).

    We note that the qualitative trends in the alkenone SST records, particularly the apparent warming in SSTs around 2.7 million years ago, are reproduced between all three datasets (Fig. 2). Therefore, while our findings do not invalidate the

conclusions of the Haug et al. (2005) study insofar as they relied upon trends in reconstructed SSTs, the amplified range of reconstructed SSTs suggest that the original SST dataset does not reflect true absolute paleo SST values.





### 3.2 A Corrected ODP Site 882 Alkenone SST Record

Correction of the 5.7 Ma alkenone SST record of Haug (1995) for methodological offsets (see **2.3**) results in an alkenone SST reconstruction from ODP Site 882 that much more closely resembles the independent SST estimates from nearby ODP

Site 883 between 3.5 to 2.6 Ma (Fig. 4). The improved agreement between the two datasets is particularly apparent when SST values from nearby samples are interpolated to facilitate point-to-point comparison (Fig. 5). After correction, the amplified range of the original ODP Site 882 SST estimates is reduced to better match the SSTs reconstructed at ODP Site 883 (Fig. 5a vs. 5b). Corrected median ODP Site 882 SST estimates more closely follow a one-to-one relationship with SST estimates from ODP Site 883 (Fig. 5a vs. 5b), but the uncertainty in the ODP Site 882 alkenone SSTs is much larger than the

ODP Site 883 SSTs because of the large residual standard error of the best fit line used for the correction (orange vs. purple shading in Fig. 4). These results suggest that our correction method removed the systematic bias in the (Haug, 1995) SST dataset that arose due to their use of the GC-CI-MS method. However, we urge caution when interpreting the SST reconstruction in samples outside the interval where it is possible to verify our correction with the data from ODP Site 883 (Fig. 4) because the differences in $C_{37:3}$Me and $C_{37:2}$Me alkenone ionization efficiency by GC-CI-MS are mass-dependent.

Meaning, we recommend that future studies be cognizant of our inability to verify whether the relationship defined in Equation 3 used to correct the ODP Site 882 record accurately captures the offset between GC-CI-MS and GC-FID $U^{K'}_{37}$ values outside of the 3.5–2.6 Ma interval where the ODP Site 883 alkenone SST record acts as an independent check on the ODP Site 882 record.

The need for caution in future interpretation of the corrected ODP Site 882 record is emphasized by the diminished but still-

present offset in maximum SST values in the corrected Site 882 alkenone SST dataset relative to the ODP Site 883 alkenone SST reconstruction (Fig. 5b). At ODP Site 883, warm alkenone SSTs broadly coincided with intervals of higher sedimentary alkenone concentrations (Novak et al., 2024). If there is a similar pattern of sedimentary alkenone concentration at ODP Site 882, which we suspect is likely given the highly correlative sedimentology between the two drill sites (Novak et al., 2024; Rea et al., 1993a, b), this would imply that our correction imperfectly accounts for the mass-dependent differences in GC-CI-

MS alkenone ionization. Nevertheless, the corrected ODP Site 882 alkenone record offers a way to utilize this important geochemical dataset in data-model comparison exercises because of both the amendment of the median SST estimates and the better quantification of the uncertainties in the underlying SST reconstruction.



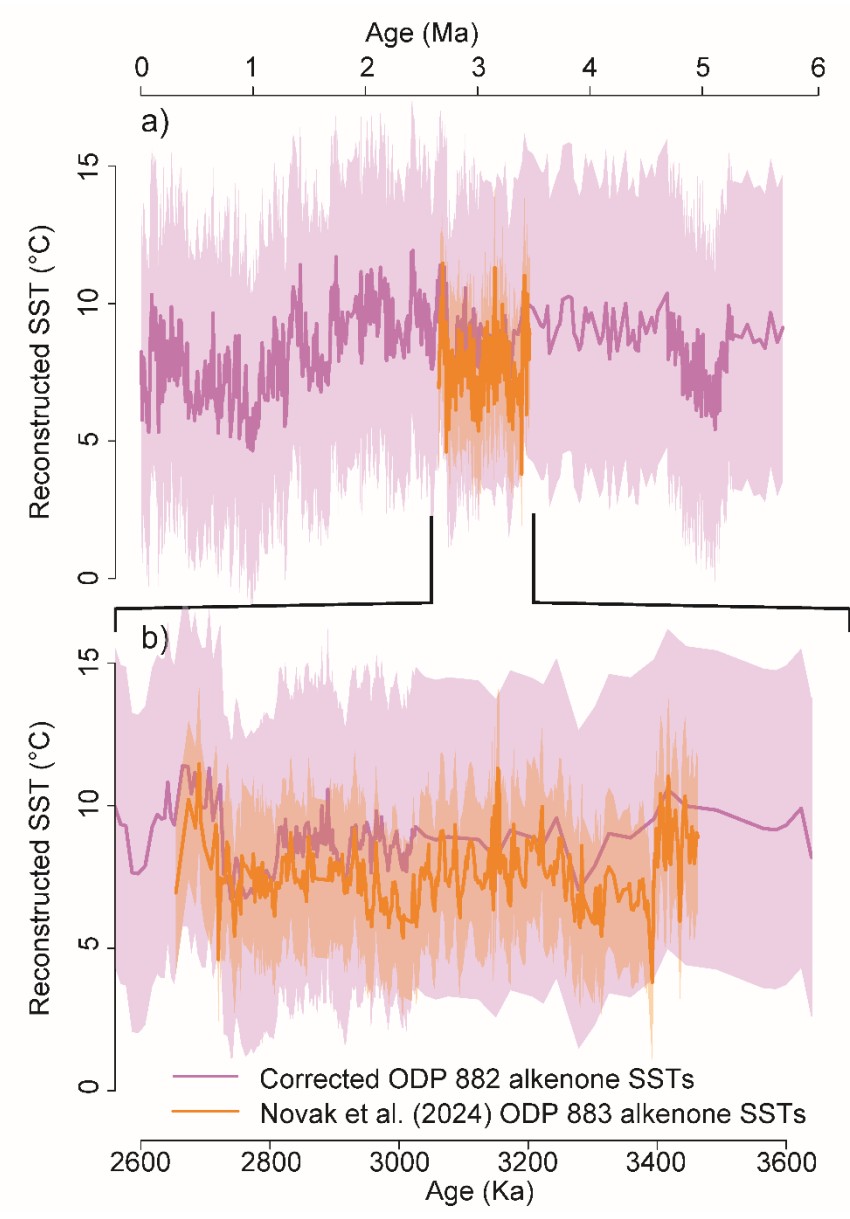

**Figure 4: Timeseries of the corrected alkenone SST record from ODP 882 compared to the nearby ODP 883 alkenone**
**SST record** (Novak et al., 2024)**. (a) the 5.7-million-year corrected ODP Site 882 alkenone SST record. (b) overlapping portion of the corrected ODP 882 alkenone SST record and the record from nearby ODP site 883** (Novak et al., 2024)**. Lines show median SST estimates. Shading shows the 95% confidence interval.**



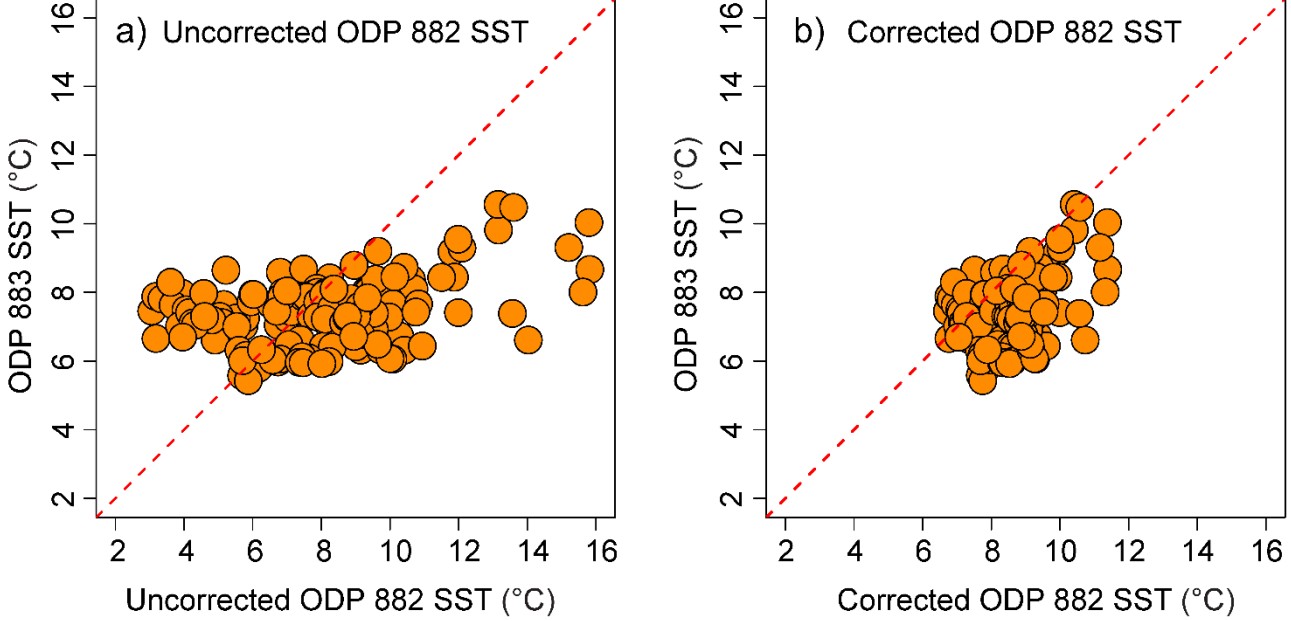

**Figure 5: Alkenone sea surface temperature reconstructions from ODP Sites 882 and 883 interpolated to show the**
**effect of correcting the ODP Site 882 SST estimates by comparison of samples of similar age. (a) before correction. (b)**
**after correction.**

## 4 Conclusions

Here, we demonstrated that the alkenone SST record from ODP Site 882 of Haug et al. (2005) is systematically offset from
other alkenone SST estimates of similar age from both the same Site and a nearby drill core (Novak et al., 2024; Studer et
al., 2012). The offsets we identified are typical of well-documented differences in the analytical techniques used by these
studies (Chaler et al., 2000, 2003). We recommend that future studies aiming to assess the accuracy of Earth System Model
simulations of Pliocene climate be cognizant of the complex offset in the absolute SST values and magnitude of SST change
reported by Haug (1995) at ODP Site 882. As an alternative to the original data, we offer a corrected dataset that accurately
accounts for the additional uncertainty in the ODP Site 882 alkenone SST estimates due to the analytical method used by
Haug (1995).



## 5 Data Availability

All alkenone data discussed in this manuscript are published and electronically archived (Haug et al., 2005; Novak et al., 2024; Studer et al., 2012). The corrected ODP Site 882 alkenone sea surface temperature record is currently in review at the PANGAEA database.

## 6 Author Contributions

JBN designed and carried out the data analysis. RPCG measured alkenones at ODP Site 883. AMG measured alkenones by GC-FID at ODP Site 882. TDH and HJD obtained funding support. JBN wrote the original manuscript draft. JBN, RPCG, TDH, HJD, and AMG edited the final manuscript.

## 7 Acknowledgements

This research used data generated from samples provided by the International Ocean Discovery Program (IODP). This work was funded by the National Science Foundation through grants 1545859 (TDH), 1459280 (TDH), and 1602331 (TDH) and the U.S. Geological Survey Ecosystems Land Change Science Program (HJD). AMG acknowledges funding from the Max Planck Society. We thank Gerald H. Haug for fruitful discussions that improved the manuscript. Any use of trade, firm, or product names is for descriptive purposes only and does not imply endorsement by the U.S. Government.

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
