# Peer review of "Correcting a Systematic Bias in an Ocean Drilling Project Site 882 Alkenone Sea Surface Temperature Record"

_EGUsphere, 2025_

## Author Comment (AC1)

**Reviewer #1 Comments**

Novak and colleagues present research that corrects for biases in previously published alkenone sea surface temperature records at ODP882. These biases stem from differences between analytical techniques for quantifying alkenone concentrations (GC-FID vs GC-CI-MS). The GC-CI-MS method enables analysis of alkenone-poor sediments that were inaccessible with GC-FID. The authors demonstrate that the original GC-CI-MS ODP882 measurements overstated climate variability, though the overall patterns and trends remain unchanged.

While the manuscript is well-written and the research methodology is sound, the manuscript's critical weakness is the absence of a meaningful discussion section. The authors successfully identify and correct the proxy bias, and highlight that the ODP882 record is very important, but fail to explore what this corrected record reveals about our understanding of Pliocene climate and Northern Hemisphere glaciation—which should be a key contribution from this methodological innovation.

Dear Reviewer #1, thank you for taking the time to review our work. We are appreciative of your time and constructive comments, which will result in an improved revised manuscript. We completely understand the criticism that our work did not include a discussion of the implications of the overstated variability in sea surface temperatures at ODP Site 882 in the original GC-CI-MS record. We will include a comparison of the original vs. corrected ODP 882 sea surface temperature dataset to mid-Pliocene model output. We expect that the most important change to the ODP 882 data will be the substantially increased uncertainty in the sea surface temperature estimates than was previously recognized, which we will highlight in a new figure.

MAJOR ISSUE: MISSING DISCUSSION OF IMPLICATIONS

The research convincingly quantifies and corrects the ODP882 bias, but stops short of addressing why this matters for the current understanding of Pliocene temperatures and Northern Hemisphere glaciation. The authors need to address fundamental questions about their findings' significance:

How does this correction alter our understanding of Pliocene temperatures and the timing/intensity of Northern Hemisphere glaciation?

What are the implications for North Pacific temperature evolution during this critical climate transition?

How might this refined record affect estimates of Earth System Sensitivity (ESS), particularly given that ODP882 is frequently cited in multi-proxy compilations?

These questions are only meant to be illustrative and motivate deeper discussion as the current manuscript provides little context for understanding the broader implications of their correction. The paleoclimate community needs to understand not just that the record was biased, but what new insights emerge from the corrected data.

Thank you for prompting the requested additional discussion with specific questions. Given that revisiting estimates of Earth System Sensitivity would be a major undertaking requiring a separate manuscript (and development of a new skillset by the authors, which does not seem realistic on the timeframe of manuscript resubmission), we will focus the new discussion of the implications of the corrected dataset for our understanding of regional sea surface temperature patterns in the late Pliocene:

1). How does the proposed correction to the ODP Site 882 manuscript impact estimates of the latitudinal temperature gradient in the North Pacific during the late Pliocene? The mid-Piacenzian Warm Period will be a particular area of focus in this new section due to the widespread community focus on the mid-Piacenzian interval for data-model comparison. The new section will include a figure comparing the latitudinal temperature gradient as estimated from paleo sea surface temperature data with the original vs. corrected ODP 882 data to PlioMIP2 model output.

2). We will compare the Western vs. Eastern subpolar North Pacific sea surface temperature estimates to understand whether the correction to the ODP Site 882 record changes our understanding of longitudinal gradient in SSTs in the late Pliocene. This analysis has relevance for observations made using Data Assimilation methods by Tierney et al. (2025) in *AGU* Advances, although we would like to emphasize that the comparison we will undertake is with the proxy data only as we lack expertise in Data Assimilation.

Warm regards and on behalf of the coauthors,

Joseph Novak

SECTION 2.1

This section could use some more text, particularly since the methods of this manuscript are intertwined with the key message: that the ODP882 record is biased and this is how you quantified and corrected for that bias. In particular, a clearer explanation of how synthetic UK37 were generated is needed for the broad paleoclimate readership of Climate of the Past.

We will add further text to explain how synthetic $U^{K'}_{37}$ values were generated to produce the black lines in Figure 3. We would like to emphasize that these synthetic

values are for illustrative purposes only and are meant to give the reader an intuitive sense of how analysis by GC-CI-MS would result in different temperature estimates as compared to GC-FID, as this is a rather technical distinction that is not easily visualized from word on the page alone. The synthetic $U^{K'}_{37}$ values are not in any way used in the corrected ODP 882 record – the corrected values arise from a simple linear regression between GC-FID and GC-CI-MS $U^{K'}_{37}$ values in proximal samples from ODP 882.

FIGURE 4

A direct comparison between the original and corrected SST records at ODP882 is conspicuously absent. Figure 4 would be the logical place to show this comparison, allowing readers to visualize both the magnitude of the correction and its impact on key climate transitions.

We will add a direct comparison to this figure. Thank you for pointing this out – it completely slipped our minds that this is an important aspect of the data to display.

MINOR COMMENT

Without demonstrating the impact of the corrected record, the manuscript somewhat overstates its importance. For instance, the abstract begins by discussing Earth climate sensitivity but many studies of Earth System Sensitivity rely on multiple records, so the authors should either: (a) demonstrate how this correction specifically affects multi-proxy compilations, or (b) focus on what unique insights about regional climate dynamics this corrected record provides.

Thank you for pointing this out, as it is not our intention to present this work as particularly "high impact." In fact, we attempt to temper the reader's impression of the impact of this work by stating in the abstract that our findings do not invalidate the conclusions regarding Earth's climate history originally drawn from the GC-CI-MS data from ODP 882. Rather, we think that the information presented here is necessary to put out for the paleoclimate science community as we strive to improve the data that we use to assess the skill of climate models at simulating sea surface temperatures under boundary conditions different from historical period.

As noted in our previous response, we will add a new section and figures to the manuscript that assesses the extent to which the corrected record improves our understanding of regional climate dynamics.

RECOMMENDATION

This manuscript makes a valuable methodological contribution by identifying and correcting an important bias in a widely used proxy from an important site. However, it currently reads more as a technical note than a full research article. To maximize its impact, the authors must add a robust discussion section that explores what this corrected record teaches us about Pliocene climate that we didn't know before. I enjoyed reading this manuscript and hope the authors find my comments useful.

Thank you for taking the time to evaluate our work.

**References Cited in Response**

Tierney, J. E., King, J., Osman, M. B., Abell, J. T., Burls, N. J., Erfani, E., Cooper, V. T., & Feng, R. (2025). Pliocene Warmth and Patterns of Climate Change Inferred From Paleoclimate Data Assimilation. *AGU Advances*, *6*(1). https://doi.org/10.1029/2024AV001356

---

## Author Comment (AC2)

**Reviewer #2 Comments**

Novak et al. present a method to correct systematic biases in UK'$_{37}$-based SST reconstructions from ODP Site 882. The original published SST estimates (Haug, 1995; Haug et al., 2005; Martínez-Garcia et al., 2010) were derived from alkenone measurements using GC-CI-MS, a method known to introduce non-linear, concentration-dependent biases. The original SST record shows systematic offsets relative to subsequent GC-FID-based records from overlapping samples at ODP Sites 882 and 883. The authors propose a linear correction approach based on the relationship between GC-CI-MS (Haug, 1995) and GC-FID (Studer et al., 2012) SST estimates where data overlap at ODP Site 882.

Dear Dr. Rattanasriampaipong,

Thank you for taking the time to evaluate our work. We are appreciative of your constructive comments and criticisms, which will result in an improved revised version of the manuscript. In particular, we will add a new section to the revised manuscript that explores the implications of our proposed correction to the ODP 882 alkenone sea surface temperature dataset for our understanding of regional sea surface temperature patterns in the Pliocene. Please find our responses to your specific comments below. Given the similarity of your comments to the criticisms raised by Reviewer #1, you may find our responses to those comments of interest as well.

Warm regards and on behalf of the coauthors,

Joseph Novak

**Major comments:**

- **Insufficient discussion of the implications and broader significance of the correction:** I echo the other reviewers' concern that the manuscript lacks adequate discussion of why this SST correction matters and how it changes our understanding of North Pacific climate evolution. The following additions would significantly strengthen the manuscript:

  We agree and will take steps to add data-model comparison and a reassessment of regional sea surface temperature gradients in the revised manuscript.

- **Data-model comparison:** The authors mention that the original SST record is unsuitable for evaluating Earth System Models but provide no actual model-data comparison. Including this comparison (either with existing model

outputs or literature values) would demonstrate the practical impact of the correction and justify the effort.

We have requested access to PlioMIP2 model output for this purpose and will include this comparison in the updated manuscript. This section will focus on how the correction (and resulting increase in the quantified uncertainty) to the ODP 882 alkenone record might change late Pliocene data-model comparison assessments of the latitudinal temperature gradient in the North Pacific basin.

- **SST gradients and ocean circulation dynamics:** The corrected SST record will substantially alter the temporal evolution of meridional temperature gradients in the northwestern Pacific. Given ODP Site 882's position within the Kuroshio Extension region, the authors should discuss implications for our understanding of western boundary current dynamics and their role in North Pacific climate variability over the past 6 Ma.

We will include a new section that reassesses the longitudinal gradient in proxy sea surface temperature estimates in the subpolar North Pacific. We will also discuss the implications of the correction for the community's understanding of the sea surface temperature variability of the Kuroshio Extension region over the past 5.7 Ma. This new discussion will focus on the reduction of the magnitude of reconstructed sea surface temperature changes that result from the proposed correction.

- **Prevalence of GC-CI-MS measurements:** The authors state that GC-CI-MS is used when alkenone concentrations are very low but cite only a handful of studies employing this method. A more systematic assessment of how commonly GC-CI-MS has been applied in the published literature would help readers evaluate the broader applicability of this correction approach. If this method has been widely used (particularly in low-productivity regions or deep-time studies), the implications extend well beyond ODP Site 882.

We conducted a literature review and will include in the introduction citations to published studies that used the GC-CI-MS technique without explicitly stating that the analytical setup accounted for the concentration-dependent ionization efficiency issues that we suspect are the reason for the issues with the ODP 882 record that we report here. These studies primarily aimed to understand paleo sea surface temperature trends through time (Durham et al., 2001; Madureira et al., 1997; Martínez-Garcia et al., 2010; McClymont et al., 2008; McClymont & Rosell-Melé, 2005; Roberts et al., 2017; Sánchez-Montes et al., 2020; Weaver et al.,

1999), but also includes a study of particulate organic matter in the modern ocean water column (Bendle & Rosell-Melé, 2004).

- **Uncertainties in extrapolating the correction beyond the calibration interval:** The linear correction is derived from a relatively brief temporal interval where the Haug (1995) and Studer et al. (2012) datasets overlap. Applying this correction to the entire 6 Ma record likely introduces additional uncertainties that must be properly quantified and propagated. Specifically:
    - The magnitude of the GC-CI-MS bias may vary systematically with alkenone concentration, sample matrix effects, or instrumental drift over time.
    - The bias may differ between glacial and interglacial periods due to changes in alkenone preservation, sediment composition, or productivity regimes.
    - The authors should provide uncertainty estimates for the corrected record and discuss how confidence in the correction degrades outside the calibration interval.

In the course of preparing our response to the reviewer comments, we came across an additional GC-FID alkenone dataset from ODP 882 (Yamamoto & Kobayashi, 2016) that expands the temporal range of the data available to attempt a correction of the Haug (1995) GC-CI-MS alkenone dataset through the entire 5.7 million year interval spanned by that record (see figure below). These new data allow us to identify more confidentially: (1) the systematic differences between the GC-FID and GC-CI-MS data from Site 882 and (2) the substantial uncertainty inherent to correcting the GC-CI-MS dataset. The revised manuscript will feature an expanded discussion of these uncertainties.

[Figure]

*$U^{K'}_{37}$ data from ODP Site 882 generated by GC-CI-MS (purple) and GC-FID (blue).*

We will also more thoroughly explore the differences between the original and proposed correction to the ODP 882 record by systematically comparing the original vs. corrected glacial vs. interglacial SST values in the Pliocene and Pleistocene through a new figure presenting box-and-whisker plots for this purpose. Lastly, we will add published Early Pliocene data from ODP Sites 883/884 (Herbert et al., 2016) to expand the timescale of the independent validation of the corrected ODP 882 record shown in Figure 4 of the original manuscript submission.

**General comments:**

- Ensure consistency in citing Haug (1995) versus Haug et al. (1995) throughout the manuscript. For example, Figure 3a shows "Haug et al. (1995)..." which appears incorrect if the reference is to a single-author 1995 publication.

Thank you for pointing this out. Yes, the proper citation here is to Haug (1995). We will correct this issue.

**Figure comments:**

**Figure 1 –** Consider adding modern SST contours or climatology to emphasize that these sites are located in the subarctic North Pacific, where accurate SST reconstructions are critical for quantifying meridional temperature gradients and evaluating climate model performance in this sensitive region.

We will add modern SST contours to the figure.

**Figure 2 –** Please extend the x-axis to cover the entire range of the data

We will extend the x-axis.

**Figure 4 –** Include a comparison showing the original versus corrected SST estimates for the entire ODP 882 record, not just a subset. This would clearly demonstrate the magnitude of the correction during different climate states (e.g., glacial-interglacial extremes) and help readers assess whether the bias is constant or varies systematically with SST.

We will add the requested comparison of the original vs corrected SST estimates from ODP 882 to Figure 4.

**Line-by-line comments:**

L17-18: Should "Haug 1995" be cited here along with "Studer et al., 2012"?

Yes – we will cite the Haug (1995) paper here also, thank you for catching that.

L18: Either remove "long" or change it to "long-chain." "Long alkenone SST" sounds odd.

Here long is intended to describe the temporal length of the record. We will replace long with the age range covered by the record to be more exact.

L65: Please provide lat/lon and water depth coordinates of ODP sites 882 and 883.

We will do this in the revised manuscript.

**References Cited in Response**

Bendle, J., & Rosell-Melé, A. (2004). Distributions of $U^{K}_{37}$ and $U^{K'}_{37}$ in the surface waters and sediments of the Nordic Seas: Implications for paleoceanography. *Geochemistry, Geophysics, Geosystems*, *5*(11), 1–19. https://doi.org/10.1029/2004GC000741

Durham, E., Maslin, M., Platzman, E., Rosell-Melé, A., Marlow, J., Leng, M., Lowry, D., & Burns, S. (2001). Reconstructingt he climatic history of the western coast of Africa over the past 1.5 m.y.: a comparison of proxy records from the Congo Basin and the Walvis Ridge and the search for evidence of the mid-Pleistocene revolution. In G. Wefer, W. H. Berger, & C. Richter (Eds.), *Proceedings of the Ocean Drilling Program, Scientific Results* (Vol. 175, pp. 1–46). http://www-odp.tamu.edu/

Haug, G. H. (1995). *Zur Paläo-Ozeanographie und Sedimentationsgeschichte im Nordwest-Pazifik während der letzten 6 Millionen Jahre (ODP-Site 882)* [Ph.D.]. Christian-Albrechts-Universität zu Kiel.

Herbert, T. D., Lawrence, K. T., Tzanova, A., Peterson, L. C., Caballero-Gill, R., & Kelly, C. S. (2016). Late Miocene global cooling and the rise of modern ecosystems. *Nature Geoscience*, *9*(11), 843–847. https://doi.org/10.1038/ngeo2813

Madureira, L. A. S., van Kreveld, S. A., Eglinton, G., Conte, M. H., Ganssen, G., van Hinte, J. E., & Ottens, J. J. (1997). Late Quaternary high-resolution biomarker and other sedimentary climate proxies in a Northeast Atlantic Core. *Paleoceanography*, *12*(2), 255–269. https://doi.org/10.1029/96PA03120

Martínez-Garcia, A., Rosell-Melé, A., McClymont, E. L., Gersonde, R., & Haug, G. H. (2010). Subpolar Link to the Emergence of the Modern Equatorial Pacific Cold Tongue. *Science*, *328*(5985), 1550–1553. https://doi.org/10.1126/science.1184480

McClymont, E. L., & Rosell-Melé, A. (2005). Links between the onset of modern Walker circulation and the mid-Pleistocene climate transition. *Geology*, *33*(5), 389. https://doi.org/10.1130/G21292.1

McClymont, E. L., Rosell-Melé, A., Haug, G. H., & Lloyd, J. M. (2008). Expansion of subarctic water masses in the North Atlantic and Pacific oceans and implications for mid-Pleistocene ice sheet growth. *Paleoceanography*, *23*(4). https://doi.org/10.1029/2008PA001622

Roberts, J., McCave, I. N., McClymont, E. L., Kender, S., Hillenbrand, C.-D., Matano, R., Hodell, D. A., & Peck, V. L. (2017). Deglacial changes in flow and frontal structure through the Drake Passage. *Earth and Planetary Science Letters*, *474*, 397–408. https://doi.org/10.1016/j.epsl.2017.07.004

Sánchez-Montes, M. L., McClymont, E. L., Lloyd, J. M., Müller, J., Cowan, E. A., & Zorzi, C. (2020). Late Pliocene Cordilleran Ice Sheet development with warm northeast Pacific sea surface temperatures. *Climate of the Past*, *16*(1), 299–313. https://doi.org/10.5194/cp-16-299-2020

Weaver, P. P. E., Chapman, M. R., Eglinton, G., Zhao, M., Rutledge, D., & Read, G. (1999). Combined coccolith, foraminiferal, and biomarker reconstruction of paleoceanographic conditions over the past 120 kyr in the northern North Atlantic (59°N, 23°W). *Paleoceanography*, *14*(3), 336–349. https://doi.org/10.1029/1999PA900009

Yamamoto, M., & Kobayashi, D. (2016). Surface ocean cooling in the subarctic North Pacific during the late Pliocene suggests an atmospheric reorganization prior to extensive Northern Hemisphere glaciation. *Deep Sea Research Part II: Topical Studies in Oceanography*, *125–126*, 177–183. https://doi.org/10.1016/j.dsr2.2015.03.005

---

## Author Comment (AC3)

Community Comment #1

**Summary**

In this manuscript Novak and co-authors use published data to propose a correction for the original alkenone-based (Uk37') sea surface temperature data from ODP Site 882 (North Pacific) that spans the Plio- and Pleistocene and was published as part of an not public PhD thesis and (partly) in (Haug et al., 2005).

The reason for this correction is that the original alkenone data was not obtained using the established GC-FID technique, but with GC-CI-MS, which could introduce a bias. For this purpose, the manuscript presents an approach based on comparing the original GC-CI-MS data with more recently published GC-FID based data from a brief Pliocene interval for Site 882 (Studer et al., 2012) as well as published GC-FID based data from a Pliocene interval from nearby Site 883 (Novak et al., 2024). The main conclusion of this manuscript is that the original SST data from Site 882 is biased, predominantly overestimating the magnitude of SST change at Site 882, but that the main conclusions of the influential (Haug et al., 2005) paper still hold.

**Main Conclusion**

The fundamental basis for this paper; namely that for the brief Pliocene interval covered by both datasets (~2750-2650 ka) the comparison between the GC-CI-MS-based SST data from Site 882 (Haug et al., 2005) with the GC-FID-based SST data from Site 882 (Studer et al., 2012) the data do not fall on the 1:1 line (e.g. Figure 3), is a valid observation. The other basis that GC-MS based approaches can lead to different UK37' and hence SSTs is also well-known (in this case I also suggest to include studies like (Hefter, 2008) into this manuscript). So there is clearly a basis that warrants a correction of the data and I appreciate the effort to correct (published) data.

However, in my opinion the approach presented in this version of the manuscript is too simplistic and needs to be more comprehensive. As such I recommend major revisions for this manuscript.

David Naafs 11th November 2025

Dear Professor Naafs,

Thank you for taking the time to evaluate our work. Your comments will result in a substantially improved revised manuscript. We are particularly appreciative of your

comments pointing out the flaws in our proposed approach to correct the ODP 882 sea surface temperature record. We outline the steps we will take in the revised manuscript to address your comments below. In particular, we would like to draw your attention to the addition of further GC-FID alkenone data from ODP 882 from (Yamamoto & Kobayashi, 2016) that broadens the "calibration" dataset for the correction to span the entire timespan of the Haug (1995) GC-CI-MS dataset. These additional data permit us to better characterize the nature of the systematic bias in the GC-CI-MS dataset and the associated uncertainties with our proposed correction to those data. Please find our specific responses to your comments below.

Warm regards and on behalf of the coauthors,

Joseph Novak

**Main Problems**

1. **Basis for (linear) correction for whole dataset is not well explained or supported by data**

The basis for the specific linear correction applied here is that the GC-CI-MS and GC-FID data for Site 882 do not fall on a 1:1 line AND that the same holds for the GC-CI-MS data from Site 882 and GC-FID data for Site 883 (shown in figure 3). However, the justification for why a linear correlation (eq. 3) is the best option to correct the data is not explained. Other options appear not explored using statistics. This while previous work suggests that the expected bias between GC-MS and GC-FID methods could be non-linear (Hefter, 2008).

Thank you for pointing out the Hefter (2008) paper to us. We were not aware of it and it provides useful further information about the nonlinear offset between GC-FID and GC-MS $U^{K'}_{37}$ values.

Our choice of a simple linear regression to correct the ODP Site 882 data was because of the distribution of the $U^{K'}_{37}$ values from Site 882 generated by GC-FID available from (Studer et al., 2012). Specifically, the Studer et al. (2012) data fall within two clusters rather than provide a continuous sampling of the full range of $U^{K'}_{37}$ values (and therefore SST estimates) in the Site 882 GC-MS $U^{K'}_{37}$ record.

The addition of the GC-CI-MS data from Yamamoto & Kobayashi (2016) substantially clarifies the nonlinear nature of the differences between the ODP 882 GC-FID and GC-CI-MS $U^{K'}_{37}$ datasets (see figure pasted below).

[Figure]

*$U^{K'}_{37}$ data from ODP Site 882 generated by GC-CI-MS vs. GC-FID.*

The comparison of the GC-CI-MS vs. GC-FID data from ODP 882 shown above closely corresponds with the comparison between the ODP 883 GC-FID $U^{K'}_{37}$ dataset and the ODP 882 GC-CI-MS $U^{K'}_{37}$ data shown in the original manuscript submission (Figure 3b); this comparison will be shown in a new supplementary figure. We will use this expanded ODP 882 GC-FID dataset as the basis of new discussion that addresses the following topics:

1). Statistical exploration of the linear vs. nonlinear relationship between the ODP Site 882 GC-CI-MS vs GC-FID $U^{K'}_{37}$ values.

2). The implications of this analysis for the shortcomings of the proposed correction to the ODP 882 dataset.

Lastly, we would like to add that we expect a likely outcome of the analysis presented in this work is that the ODP 882 alkenone dataset (both the original and our proposed correction) will no longer be used in data-model comparison exercises. We think the most important contribution of the manuscript to the literature is documenting the

issues with the ODP 882 record – the modelling community can choose to use the corrected values, or they may view the high degree of uncertainty associated with the correction as problematic. The important thing here, in our view, is that the issues with the ODP 882 record are documented for the wider community in a way that is understandable to non-specialists interested in using paleo sea surface temperature estimates to address hypotheses about past climate states.

Similarly, it is not clear why SSTs are used for this correction and not the raw UK37′ indices. It is the index that is potentially biased, the SST is just a result from that biased index. And with the use of BAYSPINE, using SSTs might introduce an additional (non-linear) bias.

This was done to simplify the error propagation since the uncertainties of Equations 3–5 are all in terms of SST. We also thought that framing the offset in terms of sea surface temperature would be easier for non-specialists to interpret since translating the $U^{K'}_{37}$ index values to a sea surface temperature is not immediately intuitive to those who do not regularly perform this calculation. In the revised manuscript, we will correct the $U^{K'}_{37}$ values rather than transform the SSTs, since the only difference is that it requires some additional calculations.

Regarding BAYSPLINE: this does not make a difference here since this calibration function is linear within the range of $U^{K'}_{37}$ / SST relevant to the ODP Site 882 record (Tierney & Tingley, 2018).

In addition, the assumption that SSTs at Site 882 should be identical to those at Site 883 during the Pliocene and across periods of major climate change (e.g. iNHG) is not well justified in the current manuscript. Present-day SSTs are the two sites are not given for reference and we know that during past climate states like the Pliocene, sites in the same ocean basin can display differences in absolute as well as SST evolution (Naafs et al., 2020).

Site 882 and 883 are both located on the Detroit Seamount ~49 nautical miles (~91 km) apart. Given their proximity and the spatial autocorrelation of sea surface temperatures on this short length scale (Hosoda & Kawamura, 2005; minimum e-folding scale of SST variability is ~1° in the Kuroshio Region), we think it is reasonable to assume that the sea surface temperature records at these two sites should be very similar to each other, at least within the uncertainties of the alkenone proxy system. We will add sea surface temperature contours to Figure 1 to better justify this assumption.

Lastly, on several occasions the statistical evidence that is needed to support statements (and importantly the correction) is lacking. For example, in lines 132-134,

the manuscript states that the SST data from Sites 883 and 882 do not appear offset and this is used to justify the correction, but no statistical evidence is given. Same for lines 153-155, stating "more closely resemble" and "improved agreement" without statistical evidence to support these claims.

We will include statistics-based assessments of the corrected ODP Site 882 dataset in the revised manuscript. This will take the form of correlation exercises, t-tests, and f-tests to assess the similarity of the corrected ODP 882 dataset to the independent dataset from ODP Site 883.

The revised manuscript needs to take these comments into account, provide a proper justification of the methods used, as well as provide statistical evidence to support the approach.

We will do so.

1. **Correction applied outside calibration range**

The entire correction for Site 882 is based on a brief Pliocene interval (~2750-2650 ka) were both a GC-CI-MS-based and GC-FID-based SST data exist. For most of the GC-CI-MS-based data that spans the last 5500 kyr, there is no GC-FID-based SST data available (outside of calibration range). Thus, the entire correction assumes that the offset remained stable across all analyses. The manuscript provides no data to support this fundamental assumption. Details are missing, but I assume that the original GC-CI-MS data were obtained across a period of time, during which MS conditions might have varied. Normally, for each batch of GC-MS runs we would run a calibration curve to correct GC-MS to GC-FID UK37' values. Hence the assumption that the correction holds across the whole record might be invalid. I wonder whether other temperature records are available, for example for during the (late) Pleistocene for Site 882 to test this hypothesis of a stable offset?

The revised manuscript needs to at least acknowledge this caveat, but ideally addresses it with other published data and/or add a couple of new GC-FID SST data from across the last 5.5 Myr from Site 882 to confirm that the offset is constant. If not properly validated, I propose to only apply the correction to the Pliocene where GC-FID data is available.

Fortunately, we found additional published GC-FID alkenone measurements from ODP Site 882 that will allow us to directly address this comment (Yamamoto & Kobayashi, 2016). These data better characterize the nonlinearity of the offset between GC-FID and GC-CI-MS $U^K_{37}$ values at ODP Site 882 (see figure in above response) and span both the

early Pliocene and Pleistocene portions of the ODP 882 record (Yamamoto & Kobayashi, 2016).

1. **Implications of corrections not clearly explained**

Assuming that following my comments in the revised manuscript the correction still holds, the authors need to expand on the implications of this correction for Plio/Pleistocene climate. Site 882 is quite an important site and the current correction leads to lower maximum SSTs and higher minimum SSTs (e.g. lines 132-134). For example, given the corrected record shown in figure 4b, the original warming across the iNHG (~2.7 Myr) that formed the foundation of the (Haug et al., 2005) paper, appears to be largely reduced (if not removed), especially when the data from around 2850 ka is taken into account.

We will take steps to discuss the implications of the proposed correction to regional reconstructions of late Pliocene climate. Please see our response to Reviewer #1 for a detailed plan of the additional sections we will add to the revised text.

We suspect that the comment about the implications of the proposed correction to the findings in the Haug et al. (2005) paper stems from our lack of a detailed discussion of the implications of the correction for a broader understanding of Plio-Pleistocene climate. Specifically, the warming feature in the ODP 882 record at 2.7 Ma was independently verified by Studer et al. (2012) (this is the GC-FID dataset we use in the proposed correction here) and is also seen at the nearby site 883 (see Figure 4c of Novak et al., 2024). We will add an additional section discussing the Plio-Pleistocene transition in the corrected record and the extent to which the shortcomings of the analytical methods of the Haug et al. (2005) paper amplified the warming signal at Site 882 across the 2.7 Ma transition.

**Minor comments:**

Line 32: both marine and terrestrial temperatures can be used for this purpose

The word "terrestrial" will be added here.

Line 50-53: also reference (Hefter, 2008) that introduces a method to use GC-MS to quantify UK37'-based SSTs (including a discussion on correcting for offset with GC-FID).

Thank you for bringing this work to our attention. We will cite the Hefter (2008) paper here also.

Line 73: justification for non-linear BAYSPINE calibration is needed

See response to previous comment. The BAYSPLINE calibration is linear in the temperature range considered here.

Line 85-90: The discussion of "instrument A/B" is not clear, revise and expand to clearly explain what this represents.

We will clarify this discussion. The intention is to make it possible for interested parties to look at the Chaler et al. (2003) paper and understand which equations we used and why.

Figure 2: the x-axis (time) stops at 2800 ka, but there is younger data shown. Make sure axis covers whole record

This will be corrected.

Figure 5: give r2 values for both panels

We understand the ask for a statistical test here, but correlation does not seem like the appropriate tool here. For example, two datasets can be correlated while not falling onto a 1:1 line, which is more so what we are interested in here. While we will report $r^2$ values, we suspect that the more important value will be whether the slope of the best fit line of the ODP 883 GC-FID $U^{K'}_{37}$ data and the corrected ODP 882 dataset approaches 1.

Line 200: Why is Prof Gerald Haug not co-author of this manuscript? It looks like the other scientists involved in creating the published Site 882 and 883 data are co-author and he was involved in discussions (line 208-209), looks weird to me. It would be a strong signal if the original author of the data is part of this correction.

We asked Prof. Haug if he would like to co-author this manuscript, but he indicated that he did not have the time to take this on given his other commitments.

**References Cited in Response**

Haug, G. H., Ganopolski, A., Sigman, D. M., Rosell-Mele, A., Swann, G. E. A., Tiedemann, R., Jaccard, S. L., Bollmann, J., Maslin, M. A., Leng, M. J., & Eglinton, G. (2005). North Pacific seasonality and the glaciation of North America 2.7 million years ago. *Nature*, *433*(7028), 821–825. https://doi.org/10.1038/nature03332

Hosoda, K., & Kawamura, H. (2005). Seasonal Variation of Space/Time Statistics of Short-Term Sea Surface Temperature Variability in the Kuroshio Region. *Journal of Oceanography*, *61*(4), 709–720. https://doi.org/10.1007/s10872-005-0078-3

Novak, J. B., Caballero-Gill, R. P., Rose, R. M., Herbert, T. D., & Dowsett, H. J. (2024). Isotopic evidence against North Pacific Deep Water formation during late Pliocene warmth. *Nature Geoscience*, *17*(8), 795–802. https://doi.org/10.1038/s41561-024-01500-7

Studer, A. S., Martínez-Garcia, A., Jaccard, S. L., Girault, F. E., Sigman, D. M., & Haug, G. H. (2012). Enhanced stratification and seasonality in the Subarctic Pacific upon Northern Hemisphere Glaciation-New evidence from diatom-bound nitrogen isotopes, alkenones and archaeal tetraethers. *Earth and Planetary Science Letters*, *351–352*, 84–94. https://doi.org/10.1016/j.epsl.2012.07.029

Tierney, J. E., & Tingley, M. P. (2018). BAYSPLINE: A New Calibration for the Alkenone Paleothermometer. *Paleoceanography and Paleoclimatology*, *33*(3), 281–301. https://doi.org/10.1002/2017PA003201

Yamamoto, M., & Kobayashi, D. (2016). Surface ocean cooling in the subarctic North Pacific during the late Pliocene suggests an atmospheric reorganization prior to extensive Northern Hemisphere glaciation. *Deep Sea Research Part II: Topical Studies in Oceanography*, *125–126*, 177–183. https://doi.org/10.1016/j.dsr2.2015.03.005